# Living-Donor Uterus Transplantation: Pre-, Intra-, and Postoperative Parameters Relevant to Surgical Success, Pregnancy, and Obstetrics with Live Births

**DOI:** 10.3390/jcm9082485

**Published:** 2020-08-03

**Authors:** Sara Yvonne Brucker, Thomas Strowitzki, Florin-Andrei Taran, Katharina Rall, Dorit Schöller, Markus Hoopmann, Melanie Henes, Martina Guthoff, Nils Heyne, Stephan Zipfel, Norbert Schäffeler, Hans Bösmüller, Falko Fend, Peter Rosenberger, Eckhard Heim, Urban Wiesing, Konstantin Nikolaou, Sabrina Fleischer, Tamam Bakchoul, Christian F. Poets, Rangmar Goelz, Cornelia Wiechers, Karl-Oliver Kagan, Bernhard Krämer, Christl Reisenauer, Ernst Oberlechner, Stephanie Hübner, Harald Abele, Pernilla Dahm-Kähler, Niclas Kvarnström, Mats Brännström, Silvio Nadalin, Diethelm Wallwiener, Alfred Königsrainer

**Affiliations:** 1Tübingen University Women’s Hospital, Calwerstr. 7, 72076 Tübingen, Germany; katharina.rall@med.uni-tuebingen.de (K.R.); dorit.schoeller@med.uni-tuebingen.de (D.S.); markus.hoopmann@med.uni-tuebingen.de (M.H.); melanie.henes@med.uni-tuebingen.de (M.H.); karl.kagan@med.uni-tuebingen.de (K.-O.K.); bernhard.kraemer@med.uni-tuebingen.de (B.K.); christl.reisenauer@med.uni-tuebingen.de (C.R.); ernst.oberlechner@med.uni-tuebingen.de (E.O.); stephanie.huebner@med.uni-tuebingen.de (S.H.); harald.abele@med.uni-tuebingen.de (H.A.); diethelm.wallwiener@med.uni-tuebingen.de (D.W.); 2Department of Gynecological Endocrinology and Reproductive Medicine, Heidelberg University Women’s Hospital, 69120 Heidelberg, Germany; thomas.strowitzki@med.uni-heidelberg.de; 3Department of Gynecology, University Hospital Zurich, 8091 Zurich, Switzerland; florin-andrei.taran@usz.ch; 4Section of Nephrology and Hypertension, Department of Diabetology, Endocrinology, and Nephrology, Tübingen University Hospital, 72076 Tübingen, Germany; martina.guthoff@med.uni-tuebingen.de (M.G.); nils.heyne@med.uni-tuebingen.de (N.H.); 5Division of Psychosomatic Medicine and Psychotherapy, Department of Internal Medicine, Tübingen University Hospital, 72076 Tübingen, Germany; stephan.zipfel@med.uni-tuebingen.de (S.Z.); norbert.schaeffeler@med.uni-tuebingen.de (N.S.); 6Institute of Pathology and Neuropathology, Tübingen University Hospital, 72076 Tübingen, Germany; hans.boesmueller@med.uni-tuebingen.de (H.B.); falko.fend@med.uni-tuebingen.de (F.F.); 7Department of Anesthesiology and Intensive Care, Tübingen University Hospital, 72076 Tübingen, Germany; peter.rosenberger@med.uni-tuebingen.de (P.R.); eckhard.heim@med.uni-tuebingen.de (E.H.); 8Department of the Ethics and History of Medicine, Tübingen University Hospital, 72074 Tübingen, Germany; urban.wiesing@uni-tuebingen.de; 9Department of Diagnostic and Interventional Radiology, University of Tübingen, 72076 Tübingen, Germany; konstantin.nikolaou@med.uni-tuebingen.de (K.N.); sabrina.fleischer@med.uni-tuebingen.de (S.F.); 10Center for Transfusion Medicine, Tübingen University Hospital, 72076 Tübingen, Germany; tamam.bakchoul@med.uni-tuebingen.de; 11Department of Neonatology, Tübingen University Children’s Hospital, 72076 Tübingen, Germany; christian-f.poets@med.uni-tuebingen.de (C.F.P.); rangmar.goelz@med.uni-tuebingen.de (R.G.); cornelia.wiechers@med.uni-tuebingen.de (C.W.); 12Department of Obstetrics and Gynecology, Sahlgrenska Academy, University of Gothenburg, 41345 Göteborg, Sweden; pernilla.dahm-kahler@vgregion.se (P.D.-K.); mats.brannstrom@obgyn.gu.se (M.B.); 13Department of Transplantation, Sahlgrenska Academy, University of Gothenburg, 41345 Göteborg, Sweden; niclas.kvarnstrom@vgregion.se; 14Stockholm IVF, 12063 Stockholm, Sweden; 15Department of General, Visceral, and Transplant Surgery, Tübingen University Hospital, 72076 Tübingen, Germany; silvio.nadalin@med.uni-tuebingen.de (S.N.); alfred.koenigsrainer@med.uni-tuebingen.de (A.K.)

**Keywords:** uterus, transplantation, infertility, Müllerian agenesis, living donor, outcome, live births

## Abstract

Uterus transplantation (UTx) can provide a route to motherhood for women with Mayer–Rokitansky–Küster–Hauser syndrome (MRKHS), a congenital disorder characterized by uterovaginal aplasia, but with functional ovaries. Based on our four successful living-donor transplantations and two resulting births, this analysis presents parameters relevant to standardizing recipient/donor selection, UTx surgery, and postoperative treatment, and their implementation in routine settings. We descriptively analyzed prospectively collected observational data from our four uterus recipients, all with MRKHS, their living donors, and the two newborns born to two recipients, including 1-year postnatal follow-ups. Analysis included only living-donor/recipient pairs with completed donor/recipient surgery. Two recipients, both requiring ovarian restimulation under immunosuppression after missed pregnancy loss in one case and no pregnancy in the other, each delivered a healthy boy by cesarean section. We conclude that parameters crucial to successful transplantation, pregnancy, and childbirth include careful selection of donor/recipient pairs, donor organ quality, meticulous surgical technique, a multidisciplinary team approach, and comprehensive follow-up. Surgery duration and blood vessel selection await further optimization, as do the choice and duration of immunosuppression, which are crucial to timing the first embryo transfer. Data need to be collected in an international registry due to the low prevalence of MRKHS.

## 1. Introduction

Uterus transplantation (UTx) has been demonstrated in recent years to provide a route to biological motherhood for women with absolute uterine factor infertility (AUFI), whose only other option to attain genetic (biological) motherhood would be surrogacy [1]. AUFI may be acquired, e.g., due to benign or malignant disease or iatrogenic loss of uterine function [1], or it may be congenital as in Mayer–Rokitansky–Küster–Hauser Syndrome (MRKHS), a rare female genital malformation occurring in approximately 4000–5000 live female births with normal female genotype [2,3]. Phenotypically, MRKHS is characterized by the congenital absence of a functional uterus and vagina in the presence of functional ovaries and external genitalia of normal appearance [3]. Also known as uterovaginal aplasia or Müllerian agenesis, amongst other terms, MRKHS occurs as type 1 or type 2, respectively defined by the absence or presence of associated malformations, mainly of the skeletal and/or the urinary system.

While human UTx is clinically still at the experimental stage, our university hospital has gained comprehensive multidisciplinary experience with MRKHS and other genital malformations over the past 25 years, offering full-range specialist services from initial diagnosis to neovaginoplasty, treatment of uterine disorders, fertility treatment, and pregnancy and delivery care [4,5,6], as well as medical care and treatment for kidney-transplanted women wishing to have children [7]. We also specialize in the treatment of emotional distress and psychosomatic disorders affecting women with MRKHS-related AUFI. This is important also with regard to the ongoing ethical debate surrounding the need for UTx as we gain a more comprehensive understanding of the emotional impact of MRKHS on patients and the potential suffering associated with unwanted childlessness. The decision to undergo UTx results from a high level of suffering due to unwanted childlessness. The potential uterus recipient and her partner, but also the potential donor, are exposed to this ongoing emotional strain. Low-threshold psychosomatic support in the form of regular contact with all three individuals involved is best suited to manage stress peaks [8,9,10].

Before the advent of UTx, the treatment of women with MRKHS focused exclusively on the vaginal aspect of the malformation, i.e., creating a neovagina by nonsurgical self-dilation or by one of the established surgical neovaginoplasty techniques [4,11]. While these treatment options enabled women with MRKHS to have vaginal intercourse, the absence of the uterus remained untreatable, preventing them from experiencing pregnancy and having their own biological children, particularly in countries such as Sweden and Germany, where surrogacy is illegal or not accepted [12,13]. Now that UTx is gradually becoming established, attention has also begun to focus on the potential impact the various neovaginoplasty techniques may have on subsequent UTx [14].

The first successful living-donor UTx procedures worldwide resulting in healthy childbirth were performed in Gothenburg, Sweden in 2012 after more than ten years of meticulous basic research, including comprehensive animal and clinical studies [12,15]. By 2019, Brännström and colleagues reported that 15 procedures had been performed in Sweden, resulting in 10 children being born from women with transplanted uteri.

We here report on the experience gained at our university hospital, the first to perform UTx in Germany in October 2016. UTx became possible at our center through close collaboration with the team at the Sahlgrenska Academy, University of Gothenburg. This involved developing a multistep process encompassing medical and psychological interventions [16]. The process developed at Tübingen University Hospital spans from clinical diagnosis, neovaginoplasty (as needed), psychological counseling, assessment of eligibility for UTx, fertility treatment, transplantation surgery, and pregnancy and delivery care, through to long-term follow-up [13,16].

The objective of the present analysis was to descriptively analyze prospectively collected observational data from uterus recipients and their living donors who underwent living-donor UTx at our hospital to identify parameters relevant to the standardization of recipient/donor selection, UTx surgery, and postoperative treatment, with the aim of implementing such parameters in a routine setting. In addition, we report our experience with the choice and duration of immunosuppressive (IS) treatment, the time of first embryo transfer (ET), treatment after a miscarriage, as well as the pregnancy, obstetric, and neonatal data for the two successful post-UTx pregnancies we had at our institution. Finally, we also provide the first-ever report of ovulation induction and oocyte retrieval under immunosuppression after UTx.

## 2. Material and Methods

### 2.1. Patients and Setting

All uterus recipients and living uterus donors who respectively completed UTx recipient and donor surgery at Tübingen University Hospital were eligible for inclusion in the present analysis. Recipients had all undergone Vecchietti-based neovaginoplasty [17,18] or created a neovagina by self-dilation. Neovaginoplasty techniques based on tissue transplants, particularly bowel segments, constituted an exclusion criterion due to the increased risk of bowel secretion-related infection during immunosuppression [14].

All potential recipients and donors had been selected according to the rigorous screening procedure reported by Taran et al. [13] and Brucker et al. [19] prior to inclusion in the Tübingen UTx program. In brief, potential recipients and donors underwent a comprehensive series of interviews, counseling, and diagnostic tests, structured as follows: (1) a pre-inclusion phase with a health questionnaire, information on program participation and surgery, and blood type and human leukocyte antigen (HLA) testing; (2) a selection phase encompassing a medical workup and clinical evaluation, consultation, diagnostic imaging, laboratory tests, and extensive psychological assessment (interview and standardized questionnaires); and (3) inclusion approval after multidisciplinary assessment of recipient and donor medical workup.

The selection phase was followed by consultation of the Regional Council Committee for Living Donation to obtain approval of living donation in accordance with the German Transplantation Act. This involved ascertaining that the donor’s decision was completely voluntary and uninfluenced by any coercive pressure, that she received adequate information about the risks and benefits involved, and that she had a close relationship with the recipient [13,19].

Uterus recipient and living-donor surgeries were performed at Tübingen University Hospital by specialists in gynecological surgery, transplantation surgery, immunology, internal medicine, and anesthesiology, who were all part of our larger multidisciplinary UTx team, which additionally included specialists in reproductive medicine, maternal-fetal medicine, neonatology, psychosomatic medicine, radiology, and pathology [13].

Figure 1 gives an overview of the screening procedures, preoperative investigations, intraoperative checkpoints, and postoperative care in the Tübingen UTx program.

### 2.2. Design and Objective

The present report was designed as a descriptive analysis of observational medical data prospectively collected from uterus recipients and their living donors during the analysis period from October 2016, when we performed the first UTx in Germany, through May 2020, by which time 1-year follow-up data were available for both babies born to two of the four uterus recipients. Specifically, parameters of interest included general pre-UTx baseline characteristics, medical and surgical history, preoperative diagnostic findings, smoking habits, fertility details, and other pre-UTx clinical characteristics. Further parameters of interest included details of the assisted reproductive technology (ART) procedure, UTx surgical report details such as surgical time, cold and warm ischemia time, surgical and postoperative complications and other events, postoperative follow-up observations, and data on post-UTx IS treatment, ET, post-UTx ART treatment, pregnancy, obstetric outcome, and neo- and postnatal findings. Surgical complications were graded based on the Clavien-Dindo (C-D) classification [20] as previously used by others to document surgical complications in living-donor uterus transplantation [21,22,23].

Our objective was to descriptively analyze relevant parameters pertaining to recipient/donor eligibility, surgery, fertility treatment, pregnancy, and later motherhood, with the ultimate goal of implementing such parameters in clinical routine to achieve standardization and ensure treatment success.

### 2.3. Ethics

In view of novel ethical challenges potentially associated with living-donor UTx, which unlike other organ transplantations is not a life-saving intervention, it was considered essential to involve the University’s Institute for the Ethics and History of Medicine from the inception of the program as described in detail by Taran et al. [13]. The Tübingen living-donor UTx clinical program received initial approval from the University of Tübingen and the Ethics Committee of the University of Tübingen (project identification code 211/2016A). All research conducted in the context of the program complied with the Declaration of Helsinki in its latest revision. All recipients and donors gave their written informed consent and provided permission to use their data for scientific research purposes prior to enrolment in the Tübingen UTx program.

### 2.4. Surgical Technique

The donor and recipient operations were performed using the standard laparotomy techniques involving subumbilical midline incisions, as previously described by Brännström et al. [12].

#### 2.4.1. Donor Surgery

Briefly, donor surgery involved harvesting the uterus with its blood vessels but without the ovaries. The Fallopian tubes were removed but not used in the graft to avoid extrauterine pregnancies. The ovaries were preserved in situ.

The arterial vasculature comprised the deep uterine artery (DUA) on either side with a segment of the internal iliac arteries (IIAs), preferably divided just distally to the gluteal artery. The venous vasculature connected to the uterine graft to be harvested comprised 1 or 2 deep uterine veins (DUVs) on either side, attached to a segment of the internal iliac vein (IIV), and/or the proximal parts of the utero-ovarian branch, divided before the inlet of the ovarian veins, in order to preserve the ovaries in situ.

#### 2.4.2. Recipient Surgery

Recipient surgery involved preparation by dissection of the vaginal vault and the external iliac vessels. Anastomoses were performed end to side from the segments of the graft’s DUAs to the recipient’s external iliac arteries (EIAs) as well as from the segments of the graft’s IIVs to the recipient’s external iliac veins (EIVs). In recipients with a thin DUV on one side, the proximal part of the utero-ovarian vein (UOV) was used for an additional venous outflow section on that side, either by anastomosis onto the segment of the graft’s IIV or directly onto the EIV. After ensuring proper reperfusion, the recipient’s vagina was opened and vaginal-vaginal anastomosis was performed. The uterus was affixed to the sacrouterine and round ligaments, and the bladder peritoneum of the graft was sutured onto the top of the recipient’s bladder.

### 2.5. Postoperative Immunosuppression

IS regimens for UTx were adopted from kidney transplantation. Immunosuppression was achieved using the following triple-drug regimens. Regimen A, the initial triple-drug IS treatment, consisted of induction therapy with anti-thymocyte globulin (ATG) at 1.5 mg/kg body weight (bw) for 3 days and the parallel start of a triple-drug IS regimen with tacrolimus (Prograf^®^ (Astellas Pharma, Munich, Germany); tacrolimus target trough level 10–12 ng/mL, starting dose 0.1 mg/kg bw), mycophenolate mofetil (MMF) 2 × 1000 mg, and prednisolone, which was tapered from 100 mg/kg bw to 5 mg/kg bw over a period of 3 weeks, followed by a maintenance prednisolone dose of 5 mg. In view of planned pregnancy, MMF in Regimen B was replaced after approximately 6 months with azathioprine (AZA) at 1 mg/kg for at least 3–6 months. Adaptation of IS treatment, necessary to improve individual tolerability, involved replacement of tacrolimus with ciclosporin (Sandimmune^®^, Novartis Pharma, Nuremberg, Germany), target trough level 100–130 ng/mL) in combination with either MMF 2 × 1000 mg or, later, when planning pregnancy, AZA 1 mg/kg bw, and prednisolone 5 mg 1-0-0.

Tacrolimus and ciclosporin trough levels were checked every 2–4 weeks, also during pregnancy, and prednisolone was retained at a 5-mg maintenance dose. Infectious prophylaxis consisted of cotrimoxazole for 6 months for *Pneumocystis jirovecii* and valganciclovir for 3–6 months, depending on the donor’s and recipient’s cytomegalovirus (CMV) status.

### 2.6. Data Collection and Analysis

Predetermined sets of medical history, diagnostic, clinical, surgical, and postoperative data were collected continually throughout the entire process from initial presentation to postoperative treatment and follow-up. Data were analyzed descriptively as numbers, frequencies, percentages, and percentiles. No statistical tests were performed.

## 3. Results

### 3.1. Recipient and Donor Clinical Characteristics at Baseline

Table 1 summarizes the baseline clinical characteristics of all uterus recipients and donors included in the analysis. Four pairs of uterus recipients and living donors underwent complete UTx procedures at our hospital between October 2016 and January 2019. Uterus Recipients 1, 3, and 4 were daughters of the respective donors; Recipient 5 was her donor’s elder sister.

Recipient age ranged from 23 to 35 years. All recipients had type 1 MRKHS. All but one recipient, who created a neovagina by self-dilation, had undergone laparoscopically assisted Vecchietti-based neovaginoplasty [17,18] to create neovaginas of 9–10 cm in length 7 to 16 years earlier.

Donor age ranged from 32 to 56 years. The 56-year-old mother of Recipient 4 was the only postmenopausal donor. All donors had given birth to at least two children, the maximum being four deliveries.

HLA matching and mismatch analyses were performed for all recipients and their respective partners, as shown in Table 1.

Excluded from the present analysis was a fifth UTx procedure (prospective Recipient 2 and Donor 2) because it was aborted after procurement of the uterus from the donor but before recipient surgery was commenced on the prospective recipient. The procedure in question was aborted based on a team decision after multiple attempts to flush the uterine arteries (UAs) of the retrieved organ during back-table preparation failed to provide adequate flow, even at high pressures, hence potentially creating a high risk of transplant failure due to inadequate perfusion after transplantation and increasing the risk that blood flow in the recipient might be insufficient to sustain uterine function, particularly during pregnancy [19]. Histopathology revealed extensive diffuse intimal fibrosis of the organ’s vascular pedicles, initial sclerosis of the right UA, and extensive intimal fibrosis in both parametric arterial segments. The decision was based on previous experience in Sweden of one case in which back-table perfusion also did not show any significant flow. The uterus was nonetheless implanted but needed to be removed again within a week, with the explanted organ exhibiting necrosis and thrombosed vessels [12]. The postoperative recovery of Donor 2 was uneventful, but she developed left-sided hydronephrosis after half a year, resulting in placement of a double J stent and re-operation 16 months after uterine procurement, involving direct ureterocystoneostomy into the left side of the bladder roof. Kidney function on the left side was reduced but stable, and overall kidney function, as determined by estimated glomerular filtration rate (eGFR), remained normal at all times. Both Donor 2 and prospective Recipient 2 developed increased emotional distress after the aborted transplantation, mainly with symptoms of depression, and sought psychosomatic support.

### 3.2. Pre-, Intra-, and Postoperative Clinical Characteristics

Table 2 summarizes the recipients’ and donors’ clinical characteristics during and after surgery as well as relevant details of the recipients’ preoperative and postoperative fertility treatments, and their pregnancies and obstetric outcomes after UTx.

#### 3.2.1. Specific Preoperative Investigations and Procedures

Uterus Evaluation

Donor uteri were evaluated using transvaginal ultrasonography and high-resolution magnetic resonance imaging of the pelvic anatomy (T2-weighted turbo spin echo (TSE)) sequences in three planes after intravenous administration of butylscopolamine bromide (Buscopan^®^, Sanofi-Aventis, Frankfurt/Main, Germany). Additionally, angiography of the lower abdomen was performed to evaluate the diameters of the arterial and venous uterine vessels and potential variants of vessel course.

Prior to transplantation, the donor underwent computed tomography (CT) angiography of the pelvic vessels with arterial and venous phase imaging to confirm presumed vessel size and exclude any new stenosis or arteriosclerotic calcifications. The presence of atherosclerotic plaques in the IIAs, small arterial caliber, and any other significant changes of the UAs were considered markers of lesser organ quality and predictors of negative outcome. Both the DUVs and the proximal part of the UOVs were assessed. The presence on either side of at least one vein that could potentially serve as an outflow was considered adequate. For recipients, a native CT scan of the pelvis was sufficient to rule out the presence of calcifying plaques in the iliac arteries.

2Ovarian Stimulation and Oocyte Retrieval

Preoperatively, all four recipients who subsequently underwent successful UTx received fertility treatment for in vitro fertilization (IVF). The numbers of oocytes fertilized with sperm from the recipients’ respective partners and subsequently cryopreserved for later implantation ranged from 9 to 17 (Table 2).

Before UTx, Recipients 1, 3, and 4 underwent ovarian stimulation for ovulation induction according to a standard antagonist protocol. Recipient 1 received daily injections of 150 IU recombinant human follicle-stimulating hormone (r-hFSH; Gonal^®^, Merck, Darmstadt, Germany), whereas Recipients 2 and 3 received daily injections of 200 IU r-hFSH from day 2 of their menstrual cycle. Pituitary suppression with cetrorelix (Cetrotide^®^, Merck, Darmstadt, Germany) started simultaneously on cycle day 6, and ovulation was induced with 250 μg recombinant human chorionic gonadotropin (r-hCG; Ovitrelle^®^, Merck, Darmstadt, Germany) when the size of the leading follicle exceeded 18 mm (day 15, day 12, or day 13). Of the 18 oocytes obtained from Recipient 1, 17 were successfully injected using the intracytoplasmic sperm injection (ICSI) technique, of which 10 were fertilized and cryopreserved as pronuclear 2 (PN-2) stage zygotes. Out of 15 oocytes retrieved from Recipient 3, 10 were injected using ICSI, of which 2 were fertilized and cryopreserved. Of the 26 oocytes retrieved from Recipient 4, 21 were injected using ICSI, 17 of which were fertilized and cryopreserved as PN zygotes.

Recipient 5 had already received ART treatment at another center in October 2015, yielding 14 cryopreserved PN zygotes.

#### 3.2.2. Intraoperative Results, Vessels Used for Anastomosis, and Intraoperative Doppler Flow Assessments

As shown in Table 2, donor surgical time ranged from 9.05 to 12.12 h in all four donors whose uteri were successfully transplanted. Estimated blood loss (EBL) was 100 mL in all donors. No donor had any intraoperative complications. However, our decision to use the right ovarian vein (OV) in Donor 3 resulted in unilateral ovariectomy. The potential use of the OVs and the consequences of this surgical alternative were preoperatively communicated to, and discussed with, the donors, all of whom provided informed consent.

Recipient surgery time ranged from 4.52 to 8.13 h, with EBL ranging from 150 to 500 mL. The recipients stayed in the intensive care unit for 2–6 days.

Total, or cold, ischemia time, i.e., from donor organ clamping to reperfusion, ranged from 111 to 175 min. Warm ischemia time, i.e., from graft placement in the recipient until organ reperfusion, thus representing part of total (cold) ischemia, was between 63 and 86 min.

Anastomosis utilized the following vessels. For arterial anastomosis, a segment of the donor’s IIA with the adjacent DUA was end-to-side anastomosed onto the recipient’s external iliac artery (EIA) on both sides. Anastomosis of the veins was performed using a segment of the donor’s IIV with the adjacent DUV. In some cases, we used the UOV, allowing the ovaries to be retained. Only in Donor 3 did we use the OV on the right side, resulting in ovariectomy on this side. Details as to which vessels were used in which case are given in Table 2.

Blood flow in the uterine arteries as determined by intraoperative Doppler ultrasonography and transit time flowmeter (Medistim, Oslo, Norway) directly after anastomosis was used to assess arterial patency and measure arterial flow, which was up to 30 mL/min in all recipients. Two Cook-Swartz Doppler Probes (Cook Medical LLC, Bloomington, IN, USA), placed on either side of the uterine artery, provided continuous audible and visual feedback of blood flow during the first 5–6 days after recipient surgery.

#### 3.2.3. Intra- and Postoperative Complications, Recovery, and Onset of Menstruation

No complications were encountered intra- or postoperatively, except in the last recipient (No. 5), who showed no arterial flow on the right side intraoperatively, thus necessitating reanastomosis. This involved suturing the right UA end to side onto the EIA in addition to a segment of the IIA with adjacent proximal vesical artery. No donor or recipient required any blood transfusions.

Hospital stays ranged from 14 to 18 days for recipients and from 11 to 14 days for donors. All four recipients and their donors had uneventful postoperative recoveries without any early or late complications during the respective follow-up periods of 4 up to 44 months. The four recipients began to menstruate between 3 and 6 weeks post UTx, and subsequently continued to menstruate regularly.

#### 3.2.4. Immunosuppression and Post-Transplantation Follow-up

Figure 2 provides an overview of the postoperative immunosuppression regimens for Recipients 1, 3, 4, and 5. It also indicates the months post UTx when changes were made to IS treatment and major pregnancy-related events occurred, including ETs, (missed) pregnancy loss, and delivery. Tacrolimus target trough levels after 3 months were 8–11 ng/mL.

All recipients started and were maintained on triple-drug Regimen A (blue in Figure 2) for the first months after UTx. Before attempting the first ET, IS needed to be switched to a triple-drug regimen without MMF for at least 3–6 months (Regimen B). This switch was made after 5 months in Recipients 1 and 3 (yellow in Figure 2). Due to elevated liver enzymes occurring after 5 months, Recipient 4 needed to be switched to ciclosporin and AZA (dark orange in Figure 2). Drug toxicity was excluded, and hepatitis E was confirmed as the cause. When the patient developed cellular rejection, in the further course, she was switched back to Regimen A in order to then routinely proceed to Regimen B after treatment of rejection and a CMV infection, as we were planning to attempt the first ET. Recipient 5 had received 3 months of Regimen A when she developed tacrolimus-associated neurotoxicity resulting in severe peripheral tremor. Consequently, calcineurin inhibitor-based IS was switched from tacrolimus to ciclosporin A for another 4 months (orange in Figure 2). Subsequently, IS was switched to the regimen without MMF (dark orange in Figure 2) for the last month of the analysis period so she would be able to proceed with her first ET after another (at least) 2 months on Regimen B.

After ET, Recipients 1 and 3 stayed on unchanged triple-drug IS therapy consisting of prolonged-release tacrolimus, AZA, and prednisolone (Figure 2).

Gynecological examinations, transvaginal ultrasound, and ectocervical biopsies were performed twice weekly during the first month after UTx and subsequently at gradually increasing intervals. Ectocervical biopsies were obtained to detect rejection and were assessed according to a uterine rejection grading scale [24].

Postoperatively, Recipients 3 and 4 each experienced one mild episode of graft rejection, which in both cases was successfully treated with a single dose of cortisone. Recipient 4 was found to have elevated liver enzymes postoperatively. Hepatitis E was confirmed and successfully treated with an antiviral therapy. Later, Recipient 4 developed a CMV infection, which was successfully treated with valganciclovir about 7 months after surgery (Table 2).

#### 3.2.5. Embryo Transfer and Postoperative Ovarian Restimulation

As indicated in Figure 2, Recipient 1 underwent two single-embryo transfers at post-UTx months 13 and 14, neither of which resulted in pregnancy. A third single-embryo transfer, which was attempted at month 17, resulted in pregnancy but ended in an early miscarriage at 8 weeks and 4 days of pregnancy. Discontinuation of progesterone resulted in spontaneous complete abortion without requiring curettage. The two remaining, preoperatively cryopreserved embryos degenerated during the fourth thawing process before ET could be attempted.

Similarly, all embryos obtained from oocytes harvested from Recipient 3 degenerated during the thawing process before ET. Therefore, postoperative restimulation became necessary in both patients.

As in preoperative ovulation induction, the following, similar, standard antagonist protocol was used postoperatively.

At post-UTx months 24 and 15, Recipients 1 and 3, respectively, received 150 IU r-hFSH (Gonal f^®^, Merck, Darmstadt, Germany) from cycle day 2 onwards. Pituitary suppression with 0.25 mg ganirelix (Orgalutran^®^, MSD Sharp & Dohme, Haar, Germany) or cetrorelix (Cetrotide^®^), respectively, started simultaneously on cycle day 6. Ovulation was induced with 250 μg r-hCG (Ovitrelle^®^) when the leading follicle exceeded 18 mm in diameter and respective estradiol levels were 1790 and 3246 pg/mL.

Ovulation induction yielded 14 oocytes for Recipient 1 and 15 oocytes for Recipient 3. Normozoospermia enabled oocytes to be fertilized using standard IVF, leading to 9 fertilized oocytes for Recipient 1 and 6 fertilized oocytes for Recipient 3. Blastocyst culture was performed with a limited number of 3 fertilized oocytes in accordance with the German Embryo Protection Act (ESchG). Supernumerary fertilized eggs were cryopreserved for future use. The luteal phase was supported by administering 600 mg progesterone daily (Utrogest^®^, Dr. Kade Besins Pharma, Berlin, Germany).

Oocytes were retrieved via the vaginal route as in other IVF procedures. No problems were encountered even though the ovaries were relocated and attached lateral to the external iliac vessels by ovariopexy during transplantation. Thus, anastomosis of the UA (and uterine vein) was performed at the level of the EIVs, resulting in both ovaries and vessels being located in a clearly cranial position than in anatomically normal IVF patients.

#### 3.2.6. Pregnancy and Obstetric Outcome

Table 2 summarizes the key findings related to pregnancy and obstetric outcome in Recipients 1 and 3.

Following the ovarian restimulation and oocyte fertilization procedures, Recipients 1 and 3 both underwent a fresh single-blastocyst transfer, resulting in successful intrauterine pregnancies. Both recipients continued triple-drug IS treatment (tacrolimus, AZA, and prednisolone) throughout pregnancy.

Recipients 1 and 3 both experienced successful pregnancies. Neither recipient had any major health problems such as diabetes or hypertension during pregnancy and childbirth. In both recipients, the umbilical artery pulsatility index was always normal. Recipient 1 was admitted to hospital 13 days before obstetric surgery (week 32 + 2) due to inguinal pain and cervical and retroplacental hypervascularization. Recipient 3 was admitted 4 days before cesarean section (week 35 + 6) due to reduced amniotic fluid and the child being diagnosed as small for gestational age (around the 10th percentile). She had previously been diagnosed with reduced cervical length (minimum length 25 mm) without contractions.

Single biopsies from both pregnant uterus recipients at week 20 of pregnancy revealed no sign of rejection. Both women gave birth to cephalically presenting healthy boys. Recipient 1 delivered her baby at week 35 + 1 after preterm rupture of membranes in the absence of any signs of infection or labor. In Recipient 3, delivery took place as scheduled. Both babies were delivered by midline-incision cesarean section.

### 3.3. Neonatal Findings and Postnatal Development

Table 3 summarizes the clinical characteristics of the babies born to Recipients 1 and 3. Respective Apgar scores were 9/10/10 and 8/8/8 and umbilical artery pH was 7.28 in both neonates. Birth weights were 2180 and 2500 g (15th percentile both), crown-heel lengths were 45 and 47 cm (15th percentile both), and head circumferences were 31 cm (8th percentile) and 31 cm (<3rd percentile), respectively.

Recipients 1 and 3 and their children were discharged from hospital 10 days and 34 days after admission and 6 days and 13 days after cesarean section, respectively. Breastfeeding began immediately in Recipient 1 and was stopped 12 months after delivery. Recipient 3 was unable to breastfeed due to previous mammillary surgery. The baby born to Recipient 1 was kept on the neonatal ward for 8 days due to premature rupture of membranes, hypoglycemia, and hypothermia. The infant subsequently developed completely normally, exhibiting bodyweights of 2.900, 6.310, 7.270, and 8.995 kg at ages 1, 4, 6, and 12 months, respectively. The child of Recipient 3 received neonatal care, including continuous positive airway pressure treatment, for three days due to preterm labor, hypothermia, and respiratory maladaptation. The infant exhibited completely normal development with bodyweights of 3.475, 5.950, 8.010, and 9.400 kg at ages 1, 4, 6, and 12 months, respectively, but required a 2-day hospital treatment for obstructive bronchiolitis at age 11 months.

### 3.4. Postpartum Menstruation and Transplant Fate

Menstruation resumed at 8 months and at 6 weeks after cesarean section in Recipients 1 and 3, respectively. The transplanted uteri were viable in all recipients as per the end of the analysis period. No events occurred that would have necessitated explantation of the graft.

## 4. Discussion

Worldwide, 54 living-donor and 19 deceased-donor UTx procedures had been performed by September 2019, resulting in the birth of 18 and 3 children, respectively, as reported at the 2nd Congress of the International Society of Uterus Transplantation (ISUTx) [25]. Our university women’s hospital was the first to perform UTx in Germany in October 2016 [16] in close collaboration with the Sahlgrenska Academy’s gynecological surgery and transplantation surgery team, who pioneered the world’s first successful human UTx [12,15]. We conducted the present analysis to summarize our clinical experience and the surgical and obstetric outcomes from the first living-donor UTx program in Germany with the aim of implementing such parameters in clinical routine to achieve standardization and ensure treatment success. To date, four uterus recipients, all with type 1 MRKHS, successfully underwent UTx with uteri from their mothers or, in one case, a sister. A fifth scheduled procedure was aborted due to poor quality of the donor organ before commencing recipient surgery, as reported previously [19]. All four women who successfully underwent UTx began to menstruate 3–6 weeks after surgery. The first two women to undergo UTx became pregnant after successful ET and each delivered a healthy baby boy.

### 4.1. Intraoperative and Postoperative Complications

EBL in our trial was 100 mL in all donors. As in the series of nine UTx procedures reported by Chmel et al. [21], no donors required blood transfusions. Furthermore, estimated donor blood loss was less than in the Dallas [23] and the Swedish study [22]. However, the median donor operative time in our study was considerably longer (10 h 40 min) than in the Czech (5.5 h) [21] and the Dallas study (6.5 h) [23] but similar to that in the Swedish study (10 h 50 min) [22].

The greatest recipient EBL was 500 mL. Blood loss in Recipient 5 was markedly higher than in all other recipients. This was due to the need for reanastomosis because the main branch of the UA was cut during an attempt to preserve the major posterior branch of the IIA on the right side in Donor 5. This was not discovered during back-table preparation as blood flowed to the uterus via collateral branches between the UA and the vesical artery. The latter proximal part had been preserved on the segment of the procured IIA. However, inadequate blood flow in the UA was discovered on the patch after revascularization. The UA was then dissected to achieve adequate length for an additional anastomosis. It was sutured end to side to the external artery in addition to a segment of the IIA with adjacent proximal vesical artery.

Adequate flow and pulsation were then detectable by flowmeter and palpation. The complication arose because the UA had the first proximal part embedded in the IIA’s vascular wall and hence the outlet was not visualized during procurement. A lesson to be learned here is that when the angiogram indicates a short distance between the first major branch and the UA, it may be preferable not to try to preserve the iliac branches on that side. If this anatomical situation is present bilaterally, which was the case in this particular donor, the risk of not leaving any proximal branch of the IIA must be weighed against the risk of revascularization failure of the UA on one side, since performing an end-to-side anastomosis between the UA and the EIA is a much more difficult task.

As UTx involves non-life-saving organ donation, only a minimal risk to the living donor is acceptable [22]. In our present study, none of the five donors experienced any severe perioperative or immediately postoperative (<30 days) complications. Hysterectomies rank among the most frequent gynecological surgery procedures, with genitourinary tract injuries occurring at a rate of 1–2% [26]. However, unlike most hysterectomies, in which the uterus is removed for benign disease and subsequently discarded, uterus donation involves far a more extensive and complex dissection of the ureters and the uterine vascular system [22]. Consequently, the rate of genitourinary tract injuries will likely be higher [22].

In our UTx program, Donor 2, whose retrieved uterus proved unsuitable for transplantation, developed hydronephrosis (C-D IIIb), presumably due to thermal injury and consecutive stricture of the ureter. Other studies reported three instances of ureteral complications occurring in living donors. Fageeh at el. [27] and Chmel et al. [21] reported two ureteric lacerations that were corrected during surgery with no further complications on follow-up. Brännström and colleagues [12] reported the postoperative diagnosis of a ureterovaginal fistula. They reimplanted the ureter approximately 4 months after uterus explantation, with no further complications occurring during follow-up. Like in the case reported by Brännström et al. [12], organ retrieval from Donor 2 was the most difficult procurement in our UTx program to date. This was due to extensive adhesions, retroperitoneal fibrosis, and a lack of retroperitoneal space owing to intra-abdominal adiposity. Nevertheless, ureteral complications are known to be surgery related and are established predictable complications after major intra-abdominal gynecological surgery procedures [12,26]. Therefore, the Tübingen follow-up protocol was changed to include both regular ultrasound examinations of the kidneys at hospital dismissal, at 4 weeks, and at 3 months after surgery to enable detection of ureteral complications due to lateral thermal spread [28], and a renal scintigram after 3 months.

### 4.2. Rejection Episodes

We observed two mild rejection episodes, one each in Recipients 3 and 4. This observation is in concord with published data [21,22]. The Czech study by Chmel et al. described 7 mild rejection episodes in 3 recipients, and one episode of severe rejection [21]. The Swedish study by Mölne and colleagues [24] reported 9 mild rejection episodes in 5 recipients, while Flyckt et al. [29] observed a single severe rejection episode in one patient. The severe rejection episodes reported by Chmel et al. [21] and Flyckt et al. [29] were managed by aggressive immunosuppressive regimens targeted at both cellular and humoral rejection and achieved complete clearance of rejection in both recipients. All reported rejection episodes, including those observed in the present study, were clinically asymptomatic and were revealed by cervical biopsies [21,22]. All mild rejection episodes were treated with cortisone [21,22]. As in the Czech study [21], the ectocervix of the two recipients with rejection episodes was inconspicuous in clinical appearance, which highlights the importance of additional regular cervical biopsies during follow-up [24].

### 4.3. ART Treatment

The present study is the first to report childbirth after implantation of a cryopreserved embryo obtained by fertilization of oocytes retrieved from a uterus recipient (Recipient 1) while under post-UTx triple-drug immunosuppression. The attempt was successful even though the ovaries were iatrogenically attached in a cranial position during transplantation. We performed IVF according to standard IVF procedures in Recipients 1 and 3. Antagonist protocols were chosen and stimulation was performed using low dosages of recombinant FSH to avoid ovarian hyperstimulation syndrome. Controlled ovarian hyperstimulation rendered the enlarged ovaries easily accessible via the standard transvaginal route. This avoided the use of alternative routes involving transabdominal or laparoscopic ovarian puncture, although these are feasible options when the ovaries are placed outside the female pelvis. Placement of the intravaginal ultrasound probe for needle aspiration of oocytes beyond the vaginal anastomosis was also feasible without difficulty, as was the exposure of the cervix for ET. Ovarian stimulation, oocyte collection and fertilization, and subsequent embryo development were unimpaired in these immunosuppressed patients. Single ETs were performed to minimize the risk of a multiple pregnancy as best possible.

According to the German IVF registry, the ongoing pregnancy rate per ET can be expected to be 33% in women under the age of 30 years [30]. Only limited data are available on immunosuppressed recipients of organ transplants and their treatment by ART. Most data are from recipients of renal and liver transplants. However, unlike uterus recipients, such vital-organ recipients have chronic diseases that adversely affect reproductive function, rendering direct comparison difficult. In a small series of renal transplant recipients, 13 patients underwent a total of 24 IVF cycles [31]. Eight women achieved 11 pregnancies. Five pregnancies ended in 3 early miscarriages before week 20 of gestation and 2 fetal losses after week 20. Six deliveries were reported without neonatal deaths. Norrman et al. investigated pregnancy outcomes in renal transplant recipients and reported normal obstetric and neonatal outcomes in a small descriptive analysis of 7 singletons and one set of twins [32]. Overall, there is evidence showing that IVF can provide a feasible route to motherhood in immunosuppressed vital-organ recipients.

### 4.4. Antenatal Complications and Obstetric Outcomes

Antenatal complications were observed in Recipients 1 and 3, who presented with preterm prelabor rupture of membranes and oligohydramnios, respectively, necessitating delivery at gestational weeks 35 + 1 and 35 + 6. Other research groups reported 3 cases of pre-eclampsia, 2 cases of cholestasis, and 1 case each of preterm prelabor rupture of membranes, pyelonephritis, and subchorionic hematoma between weeks 13 and 22 of gestation, and central placenta previa with accreta at week 21 of gestation [29,33,34,35,36].

The infants born to Recipients 1 and 3 in our UTx program both required initial neonatal care, one for 3 days due to hypoglycemia and hypothermia, and the other for 8 days due to respiratory maladaptation. However, the infants’ subsequent development was unremarkable at the routine 9- and 12-month checkups, with both children achieving catch-up growth in weight, length, and head circumference. Deliveries after UTx reported to date have all occurred between gestational week 31 + 6 and week 37, and all children were delivered by cesarean section [29,33,34,35,36]. During pregnancy, tacrolimus has so far been the immunosuppressant of choice, be it as a monotherapy or in combination with AZA and/or prednisolone. Jones and collaborators [36] point out, however, that although data in pregnant women taking tacrolimus and AZA have consistently shown that these drugs are safe to take during pregnancy with no increased risk of congenital abnormality, there is, nevertheless, an association with preterm delivery and low birthweight. However, these risks are reported to be similar across all transplant patients, irrespective of the immunosuppression, and are therefore likely to be related to maternal condition rather than treatment [36], as most patients in these studies were patients requiring renal transplantation due to renal function failure. A Swedish register-based pregnancy outcome study in 1125 women compared 980 births before and 152 after organ (66% kidney and 24% heart) transplantation and found the risk of complications to be increased after transplantation but nonetheless similar to pregnancies before transplantation [37].

### 4.5. Strengths and Limitations of the Study

This report presents relevant, hitherto unpublished original data from four patients who successfully underwent living-donor UTx for AUFI due to MRKHS, providing a detailed analysis of patient data from real-life clinical experience with UTx, including basic characteristics such as age, weight, and smoking habits, HLA matching, surgical details such as the vessels used, operative time, EBL, IS regimens, the IVF protocol, successful ovarian stimulation under IS, and data on pregnancy, childbirth, and neonatal health and development. It thus contributes substantially to the growing body of data on a pioneering, new, and continually evolving surgical technique and treatment option for women with AUFI, in particular those with MRKHS. However, the small size of our study, which is due to the low prevalence of MRKHS, may limit the overall generalizability of our results. Moreover, it should be borne in mind that living-donor UTx is clinically still at the experimental stage.

## 5. Conclusions

This report presents additional data on parameters considered relevant to the success of UTx, ART treatment, and subsequent pregnancy and childbirth. Most notably, our work has shown that it is possible to achieve and prolong pregnancies successfully under triple-drug IS treatment. Moreover, we report the standardization of triple-IS regimens with initially 5–6 months of Regimen A (tacrolimus, MMF, and prednisolone; interrupted by Regimen C (ciclosporin, AZA, and prednisolone) or switched to Regimen D (ciclosporin, MMF, and prednisolone) if necessary) followed by Regimen B (tacrolimus, AZA, and prednisolone) for maintenance immunosuppression. Of note, this is also the first study to report successful ovarian stimulation, oocyte retrieval, IVF, and ET in living-donor uterus recipients under triple-drug IS therapy, resulting in pregnancy and delivery of healthy newborns.

As previously reported, living uterus donation is associated with an estimated 20% risk of C-D grade III or grade IV surgical complications [21,22,23]. In our study, all complications were addressed and resolved, and the living donors returned to their daily activities. The rate and severity of complications are bound to decrease with growing experience in the field, as seen with living-donor kidney and liver transplantation. Reducing risk by continued evolution of the surgical technique, including the adoption of minimal access retrieval techniques, should be prioritized where possible. Following the successful live births of more than 20 infants worldwide to recipients of uteri from living and deceased donors, UTx increasingly appears to offer a viable reproductive option for women with AUFI. Furthermore, living uterus donation is well tolerated, both medically and psychologically [22]. Teams undertaking both living-donor and deceased-donor UTx are now established worldwide, and the number of UTx procedures can be expected to increase rapidly in the future.

With UTx clinically still in the experimental phase, numerous questions remain to be answered. For instance, when is the right time to remove the graft if pregnancy and childbirth are not achieved? Is a surgical curettage possible in the event of a miscarriage, and what risks would this involve? Crucially, what type of IS treatment is best given for what period of time before making the first attempt at ET? In the case of an uneventful pregnancy, when is the optimal time for delivery so as to avoid an “emergency delivery” while also preventing extremely early preterm delivery? Finally, how can surgery duration and blood vessel selection be further optimized?

Despite the continuing need for further studies to overcome the limitations of small studies, large UTx trials are unlikely to be conducted in the foreseeable future due to the relatively small numbers of eligible patients worldwide. Data therefore need to be collected in an international registry as proposed and recently initiated by the International Society of Uterus Transplantation (ISUTx).

## Figures and Tables

**Figure 1 jcm-09-02485-f001:**
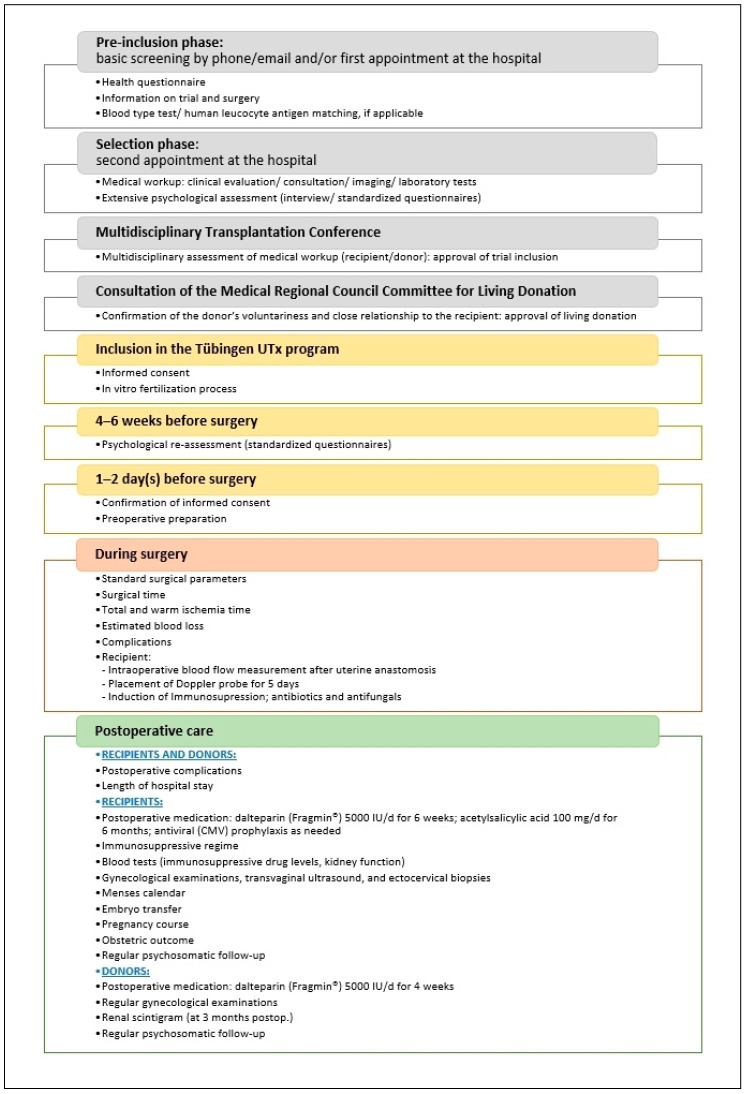
Overview of screening procedures, preoperative investigations, intraoperative checkpoints, and postoperative care in the Tübingen uterus transplantation (UTx) program (modified from [13]).

**Figure 2 jcm-09-02485-f002:**
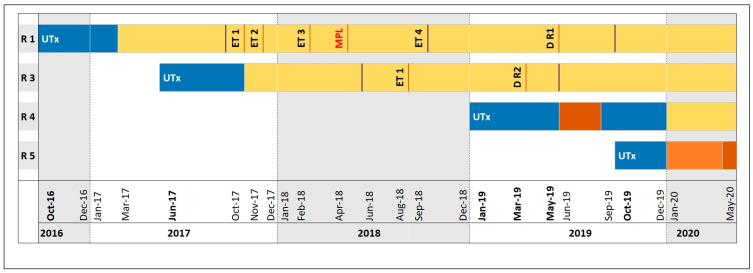
Major postoperative fertility and pregnancy related events and immunosuppressive treatment regimens in four uterus recipients. *Regimen A, blue:* induction with anti-thymocyte globulin (ATG) at 1.5 mg/kg body weight (bw) for 3 days; parallel start of a triple-drug immunosuppression (IS) regimen with tacrolimus (target trough level 10–12 ng/mL, starting dose 0.1 mg/kg bw), mycophenolate mofetil (MMF) 2× 1000 mg, and prednisolone, with prednisolone tapered from 100 mg/kg bw to 5 mg/kg bw over 3 weeks, then maintained at 5 mg/kg bw; tacrolimus target trough level after 3 months 8–11 ng/mL. *Regimen B, yellow:* triple-drug IS regimen replacing MMF with azathioprine (AZA), 1 mg/kg bw; tacrolimus target trough level checked every 2–4 weeks, also during pregnancy, and prednisolone retained at 5 mg maintenance dose. *Dark orange:* triple-drug IS with ciclosporin (80-0-80 mg, target level 100–130 ng/mL, AZA 75 mg 1-0-0, and prednisolone 5 mg 1-0-0). *Orange:* triple-drug IS with ciclosporin (100-0-100 mg, target trough level 100–130 ng/mL, MMF 2×1000 mg, and prednisolone 5 mg 1-0-0). *Abbreviations:* R, recipient; UTx, uterus transplantation; ET, embryo transfer; MPL, missed pregnancy loss at gestation week 9; D R1, delivery by Recipient 1 on 07 May 2019; D R2, delivery by Recipient 2 on 31 March 2019. Vertical black lines delimit 12-month periods of immunosuppression.

**Table 1 jcm-09-02485-t001:** Baseline clinical characteristics of recipients and donors before uterus transplantation.

	Recipient 1	Recipient 3	Recipient 4	Recipient 5	Donor 1	Donor 3	Donor 4	Donor 5
Indication for UTx	T1 MRKHS	T1 MRKHS	T1 MRKHS	T1 MRKHS	NA	NA	NA	NA
Donor-Recipient relationship	Daughter	Daughter	Daughter	Sister	Mother	Mother	Mother	Sister
Age, years	23	23	32	35	46	46	56	32
Menopausal status	Pre-	Pre-	Pre-	Pre-	Pre-	Pre-	Post-	Pre-
BMI, kg/m²	21.0	21.3	20.0	19.0	22.0	25.5	22.7	22.0
Smoking, pack years	0	5	0	0	0	5	0	0
Preoperative abstinence from nicotine, months	NA	12	NA	NA	NA	6	NA	NA
Age at neovagina creation, years	16	Self-dilation	18	19	NA	NA	NA	NA
Neovaginal length, cm	9	9	9–10	9–10	NA	NA	NA	NA
Donor’s para at baseline (type of delivery)	NA	NA	NA	NA	4 (all vaginal)	2 (both vaginal)	3 (1 cesarean, 2 vaginal)	2 (both vaginal)
Donor’s age at delivery, years	NA	NA	NA	NA	23 (Recipient 1), 24, 28, 30	22 (Recipient 3), 25	24 (Recipient 4), 27, 35	25, 27
Birthweights of donors’ children, g	NA	NA	NA	NA	3200, 3200, 3500, 3900	3600, 3800	Unknown	3085, 3460
Fetal gestational age of donors’ children at delivery, weeks	NA	NA	NA	NA	42 (all 4)	40, 41	36, 41, 41	38, 40
Blood group	A Rh−	A Rh−	O Rh+	A Rh+	A Rh+	A Rh−	O Rh+	A Rh+
Preformed donor-specific anti-HLA antibody screen	No DSA	No DSA	No DSA	No DSA	—	—	—	—
HLA mismatches	2/6 for HLA class I; 1/4 for class II	0/6 for HLA class I; 2/4 for class II	2/6 for HLA class I; 2/4 for class II	4/6 for HLA class I; 2/4 for class II	—	—	—	—
HLA mismatches with potential father	None	Not tested	None	2 repeated mismatches for HLA classes I and II	—	—	—	—
MRA: left/right uterine artery diameter, mm	NA	NA	NA	NA	2.5/2.5	3/3.5	3/3	3/3
MRA: left/right uterine vein diameter, mm	NA	NA	NA	NA	2/2	4/2	4/5	3/5

No data shown for Donor 2 or designated Recipient 2 because the UTx procedure was aborted prior to recipient surgery due to insufficient quality of the retrieved donor organ [19]. UTx, uterus transplantation; T1 MRKHS, Type 1 Mayer–Rokitansky–Küster–Hauser syndrome; NA, not applicable; BMI, body mass index; HLA, human leukocyte antigen; DSA, preformed donor-specific anti-HLA antibody; MRA, magnetic resonance angiography.

**Table 2 jcm-09-02485-t002:** Uterus transplantation: pre-, intra-, and postoperative recipient and donor clinical characteristics.^.^

	Donor 1/Recipient 1	Donor 3/Recipient 3	Donor 4/Recipient 4	Donor 5/Recipient 5
Recipient oocytes fertilized and cryopreserved for IVF preoperatively	10	6 + 2 (2 cycles of stimulation)	17	14
Recipient oocytes fertilized and cryopreserved for IVF; 2nd attempt, postoperative	9	6	NA	NA
UTx, month/year	10/2016	06/2017	01/2019	10/2019
Surgical time for donor/recipient, hours	12.12/5.95	9.05/4.52	10.40/6.20	9.19/8.13
Blood vessels used for anastomosis	Left side: – DUA (D) E/S onto EIA (R) – DUV (D) E/S onto EIV (R) Right side: – DUA (D) E/S onto EIA (R) – DUV (D) E/S onto EIV (R)	Left side: – DUA (D) E/S onto EIA (R) – DUV (D) E/S onto EIV (R) Right side: – DUA (D) E/S onto EIA (R) – OV (D) E/S onto EIV (R) with unilateral (right) ovariectomy	Left side: – DUA (D) E/S onto EIA (R) – DUV (D) and uterine branch of UOV (D), both E/S onto EIV (R) (UOV cranially from DUV) Right side: – DUA (D) E/S onto EIA (R) – DUV (D) E/S onto EIV (R)	Left side: – DUA (D) E/S onto EIA (R) – Uterine branch of UOV (D) with anastomosis onto DUV (D) onto EIV (R) Right side: – DUA (D) onto EIA(R) – DUV (D) and uterine branch of UOV (D) both E/S onto EIV (R) (UOV cranially from DUV)
Total ischemia time ^1^, min	111	119	153	175
Warm ischemia time ^2^, min	63	77	86	83
Estimated blood loss in donor/recipient, mL	100/200	100/150	100/200	100/500
Surgical complications in donor/recipient	None/None	None/None	None/None	None/Intraoperative reanastomosis of right DUV
Length of hospital stay of donor/recipient, days	11/18	12/17	14/ 14	14/15
Recipient’s first menstruation, weeks post UTx (only)	6	6	3	5
Graft rejection by recipient, treatment	None	1 mild episode, successfully treated with 1 cortisone pulse over 3 days	1 mild episode, successfully treated with 1 cortisone pulse over 3 days	None
Other postoperative events in donor/recipient	None/None	None/None	Donor: none/ Recipient: elevated liver enzymes confirmed hepatitis E, successful antiviral therapy; CMV infection, successfully treated with valganciclovir	None/None
Recipient pregnancies after UTx	2; 1 missed pregnancy loss at gestation week 8	1	NA	NA
Recipient deliveries after UTx	1	1	NA	NA
Recipient’s mode of delivery	Secondary cesarean section	Primary cesarean section	NA	NA
Time from incision to recipient’s delivery, min	8	20	NA	NA
Overall recipient surgery time for delivery, min	59	70	NA	NA
Recipient’s age at delivery, years	26	25	NA	NA
Gestational week + days at delivery	35 + 1 after preterm prelabor rupture of membranes	36 + 3, mild oligohydramnios	NA	NA
Placental histology	No pathology	No pathology	NA	NA
Explantation of the transplanted uterus	No	No	No	No

No data shown for Donor 2 or designated Recipient 2 because the UTx procedure was aborted prior to recipient surgery due to insufficient quality of the retrieved donor organ [19]. ^1^ Total ischemia time = cold ischemia time, i.e., the time from donor organ clamping to reperfusion. ^2^ Warm ischemia time = time from graft placement in the recipient until reperfusion; warm ischemia time represents part of total (= cold) ischemia time. IVF, in vitro fertilization; NA, not applicable; UTx, uterus transplantation; DUA, deep uterine artery with internal iliac artery (IIA) segment; E/S, end-to side anastomosis; D, donor; EIA, external iliac artery; R, recipient; DUV, deep uterine vein with internal iliac vein (IIV) segment; UOV, utero-ovarian vein; OV, ovarian vein; EIV, external iliac vein; CMV, cytomegalovirus.

**Table 3 jcm-09-02485-t003:** Clinical characteristics of two newborns from two women with transplanted uteri.

	Baby Born to Recipient 1	Baby Born to Recipient 3
Baby’s sex	Male	Male
Month/Year of birth	05/2019	03/2019
Presentation	Cephalic	Cephalic
Gestational age at birth, weeks + days	35 + 1	36 + 3
Birthweight, g (percentile)	2180 (15th)	2500 (15th)
Crown-heel length at birth, cm (percentile)	45.0 (15th)	47.0 (15th)
Head circumference at birth, cm (percentile)	31.0 (8th)	31.0 (< 3rd)
Neonatal health status	Healthy	Healthy
Apgar score at 1/5/10 min	9/10/10	8/8/8
Umbilical artery blood pH	7.28	7.28
Blood group	A Rh−	A Rh−
Diagnoses	Neonatal hypoglycemia and hypothermia	Respiratory maladaptation, CMV negative
Treatment	Early feeding, warming bed	CPAP
Bodyweight at hospital discharge, g (percentile)	2376 (5th)	2370 (4th)
Crown-heel length at hospital discharge, cm (percentile)	45.0 (< 3rd)	47.0 (8th)
Head circumference at hospital discharge, g (percentile)	31.5 (< 3rd)	31.0 (< 3rd)
Bodyweight at age 6 months, g (percentile)	7270 (25th)	8010 (48th)
Crown-heel length at age 6 months, cm (percentile)	63 (5th)	70 (52nd)
Head circumference at age 6 months, g (percentile)	42 (12th)	43.5 (50th)
Bodyweight at age 12 months, g (percentile)	8995 (25th)	9300 (27th)
Crown-heel length at age 12 months, cm (percentile)	72.5 (11th)	75 (50th)
Head circumference at age 12 months, g (percentile)	47 (50th)	46.5 (48th)

CPAP, continuous positive airway pressure; CMV, cytomegalovirus.

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
