# Peer review of "Living-Donor Uterus Transplantation: Pre-, Intra-, and Postoperative Parameters Relevant to Surgical Success, Pregnancy, and Obstetrics with Live Births"

_jcm, 2020, doi:10.3390/jcm9082485_

Round 1

Reviewer 1 Report

Dear authors

Thank you for this interesting and comprehensive article which contributes to the improvement of knowledge about UT.

 I am in favour of a publication

However minor clarifications are nececessary:

  • The introduction should be reduced
  • In methodolgy, it lacks artery anastomosis description (around line 200)
  • Results:
    • could you describe type of neovaginoplasty. Digestive flap are usually not recommended in  UT trials (around line 240)
    • in table 2
      • it is not correct to consider no complication in case 5 (reanastomosis equired)
      • the Recipient 2 with hydronephrosis complication should appear in this table
      • terms as"secondary "or "primary " cesarean don't seem adequate
    • the ovariectomy in donor 3 should be considered as complication (line 341)
    • Do the patient have antenatal coricosteroids injection to prevent prematurity ?
    • No explantation is discussed. What is the protcol decision or the choice of patient about additional pregnancy ?

Author Response

Please find our responses in the attached PDF file.

Thank you.

Reviewer 2 Report

The authors Brucker et al present a detailed case series of UTx (four cases) from prospectively collated data. This work focuses on women with MRKHS. In this work the authors outline the clinical care, pre-operative parameters, operative technique, post operative complications and fertility outcomes. This work is comprehensive. The authors should be commended for both their outcomes and adding these data to the scientific literature. 

Major concerns: None

Minor concerns: 

Overall comment: This manuscript, though well written, is dense and at times hard to distil. Although key parameters are presented in tabular form, I feel some extra figures outlining their key learning points could be presented so as to draw the key facts from the text.

If I may, I would suggest a flow diagram outlining the clinical pathway for patients starting with pre-counselling and assessment, relevant pre-surgical laboratory investigations/imaging, key intra-operative checkpoints and post operative care.

In addition, a box of bullet point key points/lessons learned would be helpful. Furthermore, outlining what contra-indications they have for UTx for both the donor and the recipient. 

Please could the authors ensure that in the discussion they limit their findings to women with MRKHS and highlight that generalising to UTx outside of that population, on the basis of their data/experience is problematic. 

Could the authors also offer an explanation as to why their intra-operative times where so different to the Czech study - was it their patient group were more surgically challenging? 

In the text, two appendices are mentioned (A and B - after CoI) - I cannot see these. 

Thank you for pressing this fascinating work. 

Author Response

(The authors gave the same response as above.)

Reviewer 3 Report

Your paper il very interesting and original.

In my opinion could be useful a sort description of died donor tecnique.

Author Response

(The authors gave the same response as above.)
